# Simultaneous Detection of *Salmonella* spp. and Quantification of *Campylobacter* spp. in a Real-Time Duplex PCR: Myth or Reality?

**DOI:** 10.3390/pathogens12020338

**Published:** 2023-02-16

**Authors:** Nagham Anis, Laetitia Bonifait, Ségolène Quesne, Louise Baugé, Marianne Chemaly, Muriel Guyard-Nicodème

**Affiliations:** Unit for Hygiene and Quality of Poultry and Pork Products, Laboratory of Ploufragan-Plouzané-Niort, ANSES, 22440 Ploufragan, France

**Keywords:** foodborne pathogen, poultry, process hygiene criterion, qPCR

## Abstract

In Europe, there is a process hygiene criterion for *Salmonella* and *Campylobacter* on broiler carcasses after chilling. The criterion gives indicative contamination values above which corrective actions are required by food business operators. The reference methods for verifying compliance with the criterion for *Salmonella* and *Campylobacter* are international standards EN ISO 6579-1 (2017) and EN ISO 10272-2 (2017), respectively. These methods are time-consuming and expensive for food business operators. Therefore, it would be advantageous to simultaneously detect *Salmonella* spp. and quantify *Campylobacter* in the same analysis, using the same sample after the pre-enrichment step for *Salmonella* recovery. A duplex PCR for *Salmonella* detection and *Campylobacter* spp. enumeration was developed. Considering the method as a whole, the LOD and LOQ for *Campylobacter* enumeration were slightly over the limit of 3 log CFU/g set by the process hygiene criterion. A comparison of the duplex PCR method developed with the ISO method on artificially contaminated bacterial suspensions and on naturally contaminated samples demonstrated a good correlation of the results for *Campylobacter* enumeration when the duplex PCR was performed on samples taken before or after the pre-enrichment step, but revealed a slight bias with a large standard deviation resulting in widely spaced limits of agreement.

## 1. Introduction

*Campylobacter* spp. and *Salmonella* spp. are the leading bacterial pathogens causing gastroenteritis in humans. In Europe, during the year 2020, 120,946 cases of campylobacteriosis and 52,702 cases of salmonellosis were reported [1]. Even though the number of reported cases for these zoonoses was lower than in previous years due to the COVID-19 pandemic and the withdrawal of the United Kingdom from the EU, they remained the most frequently reported zoonotic diseases [2]. Poultry is generally recognized as the main source of infection of these two bacteria [3,4,5].

Commission Regulation (EC) No. 2073/2005 [6] (https://eur-lex.europa.eu/legal-content/EN/TXT/PDF/?uri=CELEX:32005R2073&from=fr, accessed on 15 January 2023) has established microbiological criteria for foodstuffs. In particular, there is a process hygiene criterion (PHC) for *Salmonella* on broiler carcasses after chilling. This regulation was amended in 2018 to include *Campylobacter* according to Commission Regulation (EU) No. 2017/1495 [7] (https://eur-lex.europa.eu/legal-content/EN/TXT/PDF/?uri=CELEX:32017R1495&from=FR, accessed on 15 January 2023). The regulation sets indicative contamination values above which corrective actions are required by food business operators.

Results are considered satisfactory if *Salmonella* spp. are detected in a maximum of 7 out of 50 samples. However, in France, a stricter limit for *Salmonella* is set, i.e., 5 out of 50 samples (Instruction technique DGAL/SDSSA/2018-23 09/01/2018; https://info.agriculture.gouv.fr/gedei/site/bo-agri/instruction-2018-23, accessed on 15 January 2023) [8]. For *Campylobacter* spp., results are considered satisfactory if a maximum of 15 out of 50 samples have a *Campylobacter* enumeration over 1000 CFU/g. The PHC for *Campylobacter* will gradually become more stringent over time and the maximum number of samples allowed with this level of *Campylobacter* will be 10 in 2025. In the event of unsatisfactory results, an improvement in slaughter hygiene, a review of process controls, and on-farm biosecurity measures are required.

The reference methods used for verifying compliance with the criteria for *Salmonella* and *Campylobacter* in poultry carcasses are international standards EN ISO 6579-1 [9] and ISO 10272-2 [10], respectively. The method for *Salmonella* detection follows a standard protocol of non-selective pre-enrichment, followed by selective enrichment, isolation on selective agar media, and finally, biochemical and serological confirmation. The method for *Campylobacter* enumeration requires dilution and plating on selective agar media, isolation on agar media, and finally, biochemical confirmation. These methodologies are time-consuming (they take about 4–5 days) and require specific culture conditions (microaerobic atmosphere) for *Campylobacter*.

The sampling plan for the PHC on *Campylobacter* follows the same testing approach as for *Salmonella*, i.e., as recommended in Regulation (EC) No. 2073/2005 [6]. Therefore, the recommended neck skin samples may be used for testing compliance with both PHCs. The ultimate objective would thus be to develop a molecular method for real-time PCR enabling the simultaneous detection of *Salmonella* spp. and quantification of *Campylobacter* spp. in the same dual-purpose analysis that would lower both costs and the time needed. The present work was initiated in order to test the proof of concept that *Salmonella* could be detected and *Campylobacter* enumerated from the same sample of broiler neck skins and with a single reaction, assuming that *Campylobacter* enumeration by PCR would not be impaired after the pre-enrichment step needed for *Salmonella* detection. To verify this hypothesis, several prerequisites have to be fulfilled, which is the objective of this study. First, a duplex PCR enabling the concomitant specific detection of *Salmonella* and *Campylobacter* needs to be implemented. After, it should be demonstrated that *Salmonella* pre-enrichment is not impaired by co-incubation with *Campylobacter*. Moreover, while a previous study has already demonstrated that *Campylobacter* does not grow during this pre-enrichment step [11], it should be established that enumeration by qPCR is not impaired after the pre-enrichment step. Finally, there should be a comparison between the duplex PCR method and the ISO method on artificially contaminated bacterial suspensions and on naturally contaminated samples.

## 2. Materials and Methods

### 2.1. Strains and Culture Conditions

The bacterial strains used in this study are listed in Table 1: 27 *Campylobacter* strains, with different *C. jejuni* genotypes determined by multi-locus sequence typing (MLST); 26 different *Salmonella* serovars; and 13 other bacterial species were tested for the specificity of the duplex PCR.

*Campylobacter* strains were subcultured from stock solutions stored at −80 °C by cultivating them on Columbia blood agar (Thermo Fisher Diagnostics, Dardilly, France) and modified charcoal–cefoperazone–deoxycholate agar (mCCDA, Thermo Fisher Diagnostics, Dardilly, France). All the plates were incubated at 41.5 °C for 48 h under microaerobic conditions (5% O_2_, 10% CO_2_ and 85% N_2_) in a jar equipped with a microaerobic atmosphere generating kit (Whitley jar gassing system; Labo and CO, Marolles-En-Brie, France) or with a CampyGen sachet (Thermo Scientific, Tokyo, Japan). One single typical colony of *Campylobacter* was chosen from the mCCDA to inoculate 5 mL of BB (brucella broth , Becton, Dickinson and Company, Le Pont-de-Claix, France). After microaerobic incubation at 41.5 °C for 24 h, 100 µL was added to another 5 mL of BB and incubated microaerobically at 41.5 °C for 18 h. Next, 1 mL was centrifuged at 13,000× *g* for 5 min and the pellet was used for DNA extraction. *Salmonella* and other bacterial strains were subcultured from stock solutions stored at –80 °C by cultivating them on plate count agar (PCA, bioMérieux, Craponne, France). All the plates were incubated at 37 °C for 24 h under aerobic conditions. One single characteristic colony of *Salmonella* was chosen from the PCA to inoculate 5 mL of BHI (brain heart infusion) broth (Biokar Diagnostics, Allonne, France). After aerobic incubation at 37 °C for 24 h, 1 mL was centrifuged at 13,000× *g* for 5 min and the pellet was used for DNA extraction.

### 2.2. DNA Extraction

DNA for specificity testing and performance characteristics determination of the duplex PCR was extracted using a mericon™ DNA Bacteria Kit (Qiagen, Courtaboeuf, France). The QIAamp^®^ DNA mini kit (Qiagen, Courtaboeuf, France) was used for DNA extraction from the broiler neck skin samples. Both kits were used according to the manufacturer’s recommendations.

### 2.3. Real-Time PCR

The primers and probes (Merck Darmstadt, Germany) used in this work have already been described by Lund et al. (2004) [12] and Malorny et al. (2004) [13] (Table 2). The reporter dye was replaced with HEX for the probe targeting the *Campylobacter* in order to facilitate differentiation between two pathogens in a single reaction. Each assay was performed in a total volume of 20 µL containing 10 µL of PerfeCTa^®^ qPCR ToughMix^®^ (Quantabio, Beverly, MA, USA), 900 nM of *Campylobacter* primers, 100 nM of *Salmonella* primers, 125 nM of each probe, and 2 µL of DNA template. PCR amplifications were performed in the CFX96 thermal cycler (Bio-Rad, Marne-La Coquette, France) as follows: initial denaturation at 95 °C, 10 min followed by 40 cycles of 15 s at 95 °C and 1 min at 60 °C. Each sample was run in duplicate, and each run included positive (genomic DNA of each target pathogen) and negative (no template control) controls.

### 2.4. Evaluation of the Duplex PCR’s Specificity and Performance Characteristics

The specificity of the duplex PCR was evaluated for *Campylobacter* and *Salmonella* strains described in Table 1. PCR amplifications were performed in duplicate as described in Section 2.3.

Both the performance characteristics of the duplex PCR and the determination of its LOD and LOQ were evaluated using genomic DNA of *C. jejuni* Anses640 and *S*. Blegdam 421, prepared as described in Section 2.2. Two replicates of ten independent runs of serially diluted genomic DNA were analyzed. The initial concentration of the bacterial suspensions used to prepare the genomic DNA was determined after a plate count showing that it contained 3.36 × 10^8^ CFU/mL of *C. jejuni* Anses640 and 8.80 × 10^8^ CFU/mL of *S*. Blegdam 421. DNA was extracted as described in the previous paragraphs. The genomic DNA of the standard cultures was extracted as mentioned in Section 2.2 and serially diluted. The bacterial load of *Campylobacter* and *Salmonella* ranged from 0.23 to 6.23 log CFU and from 0.64 to 6.64 log CFU, respectively.

### 2.5. Comparison of the Duplex PCR Method with the ISO Methods on Artificially Contaminated Bacterial Suspensions

The assays were performed following the procedure described in Figure 1. *C. jejuni* Anses640 was cultured in 5 mL of BB as described in Section 3.4. Serial dilutions (1:10 *v*/*v*) were then carried out to inoculate bags containing 250 mL BPW (buffered peptone broth, bioMerieux, Marcy-l’Etoile, France) with *C. jejuni* at different final concentrations ranging from 2 to 6 log CFU/mL. *S*. Blegdam was used directly from frozen aliquots containing 10 CFU/mL preserved in glycerol peptone broth and stored at –80 °C. These aliquots were thawed and added to BPW bags. Then, 10 mL of inoculated BPW was collected for *Campylobacter* spp. enumeration according to the EN ISO 10272-2 method [10] and 1 mL for *Campylobacter* spp. for DNA extraction and enumeration using the duplex PCR (Figure 1). The remaining inoculated BPW was incubated for pre-enrichment of *Salmonella* spp. in aerobic conditions at 37 °C for 16 h. After incubation, 1 mL of the inoculated BPW was collected for DNA extraction, followed by *Campylobacter* spp. enumeration and *Salmonella* spp. detection using the duplex PCR. The presence of *Salmonella* spp. was checked according to EN ISO 6579-1 [9] using the remaining inoculated BPW (Figure 1).

### 2.6. Evaluation of the Duplex PCR Method on Naturally Contaminated Neck Skin Samples

To evaluate the diagnostic potential of the developed duplex PCR, naturally contaminated broiler neck skin samples were collected from a French slaughterhouse and transported in cold conditions to the laboratory. Three neck skin pieces (about 10 g each) from the same batch were pooled to constitute a sample unit of at least 25 g. Twenty units from five different batches were analyzed: 25 g of neck skin samples was homogenized in 250 mL of BPW (1:10 (*m*/*v*)). Then, 10 mL of the suspension was collected for *Campylobacter* spp. enumeration according to the EN ISO 10272-2 method [10], and 1 mL was collected for *Campylobacter* spp. for DNA extraction and enumeration using the duplex PCR. The suspension was then incubated for pre-enrichment of *Salmonella* spp. in aerobic conditions at 37 °C for 16 h. After incubation, 1 mL of the suspension was collected for DNA extraction, followed by *Campylobacter* spp. enumeration and *Salmonella* spp. detection using the duplex PCR, and the presence of *Salmonella* spp. was checked in parallel according to EN ISO 6579-1 [9] using the remaining suspension.

### 2.7. Statistical Analysis

Data were log10-transformed to ensure that they were normally distributed. A correlation analysis and Bland–Altman plots [14] were executed using GraphPad Prism version 5.0 (GraphPad Software, La Jolla, CA, USA). *p* < 0.05 was considered statistically significant.

## 3. Results

### 3.1. Performance Efficiency of the Developed Duplex PCR

The duplex PCR, evaluated using serial dilutions of the target pathogens *Campylobacter* and *Salmonella,* showed a linear relationship between the DNA input (log CFU) and the threshold cycle (Cq) value for both targets (Figure 2). An amplification efficiency of 99.7 ± 1.5% and 97.7 ± 2.4% was obtained for *Campylobacter* and *Salmonella*, respectively (Table 3).

### 3.2. Specificity and Sensitivity of the Duplex PCR

The specificity of the duplex PCR was evaluated in this work against a panel of 27 *Campylobacter* strains with different *C. jejuni* genotypes, 26 *Salmonella* serovars, and 13 other bacterial species. The duplex PCR detected *Campylobacter* strains and all the tested *Salmonella* serovars. No signal was observed for any other bacterial species tested. The results obtained using the duplex PCR comply with the expected results (data not shown). In this study, the lowest amount of DNA standard for *Campylobacter* and *Salmonella* gave a positive result in all the replicates, so it was considered the LOD for this PCR. The LOD for *Campylobacter* was 0.23 log CFU per reaction and 0.64 log CFU per reaction for *Salmonella*. The limit of quantification (LOQ) determined for *Campylobacter* is shown in Table 4. The results presented in Table 4 reveal that the LOQ of *Campylobacter* is 1.23 log CFU/reaction with a coefficient of variation of 4.83%, which is the acceptable level. The highest variability (67.70%) was observed in the reaction having the lowest quantity of *Campylobacter*, i.e., 0.23 log CFU/reaction (Table 4).

### 3.3. Evaluation of the Duplex PCR Method to Detect Salmonella in the Presence of Campylobacter

As shown in Table 5, *Salmonella* was detected and similar PCR amplification was observed after the pre-enrichment step when co-incubated with different concentrations of *C. jejuni*. Thus, *S*. Blegdam was enriched and detected by the duplex PCR, regardless of *Campylobacter* concentration.

### 3.4. Comparison of Campylobacter spp. Counts Obtained by the Duplex PCR Method before and after the Pre-Enrichment Step with the Microbiological Method (EN ISO 10272-2)

Considering the duplex PCR’s LOQ for *Campylobacter* enumeration (1.23 log CFU per reaction), assuming no loss during DNA extraction and taking into account the elution volume, the LOQ corresponded to 3.23 log CFU/mL. Figure 3 presents the correlation of the results obtained by duplex PCR before and after the pre-enrichment step with the results obtained with the microbiological method. All the samples were positive with the duplex PCR, but several results were below the LOQ (Figure 3). It was decided to keep these results for further analysis. A correlation analysis, performed using Pearson’s correlation coefficient, demonstrated a strong positive linear relationship between the results obtained by the duplex PCR method and the microbiological method either before (Pearson correlation coefficient = 0.945; *p* < 0.001) or after (Pearson’s correlation coefficient = 0.960; *p* < 0.001) the pre-enrichment, as a value of 1 for this coefficient indicates a perfect linear relationship (Figure 3).

Moreover, Bland–Altman plots were constructed to assess the agreement between the microbiological method and the duplex PCR method before and after the pre-enrichment step (Figure 4). All the tested samples were within the 95% confidence interval limits (±1.96 SD); however, 1.96 SD values were relatively high, ranging from to 0.85 log CFU/mL (after pre-enrichment) to 0.97 log CFU/mL (before pre-enrichment). A slight and non-significant bias (representing the mean difference between the methods) towards underestimation of the qPCR method before (−0.19 log CFU/mL) and after (−0.17 log CFU/mL) pre-enrichment was observed (Figure 4).

### 3.5. Evaluation of the Duplex PCR Method to Detect Salmonella spp. and Quantify Campylobacter spp. on Naturally Contaminated Broiler Neck Skins Compared with the Microbiological Method (EN ISO 6579-1 and EN ISO 10272-2)

Twenty samples (pools of three neck skins) were analyzed following the EN ISO 6579-1 [9] and EN ISO 10272-2 [10] methods. DNA was extracted using the qiaAmp kit because poor results were obtained using the Mericon kit (data not shown). The duplex PCR was performed on these samples before and after the pre-enrichment step. The results are presented in Figure 5. The presence of *Salmonella* spp. was detected in one sample (neck skin sample no. 20) with the microbiological method and the duplex PCR after enrichment. *Campylobacter* was enumerated in 19 samples (except neck skin sample no. 1) with the microbiological method and in 20 samples with the duplex PCR before and after the pre-enrichment step (Figure 5). As shown in Figure 5, thirteen samples presented enumeration over the limit of 3 log CFU/g set by the PHC for *Campylobacter* using the microbiological method. Among them, 85% (11/13 samples) and 70% (9/13 samples) of the samples were also enumerated accordingly by the duplex PCR before and after the pre-enrichment step, respectively (Figure 5). In the same way, 71% (5/7 samples) and 86% (6/7 samples) with a *Campylobacter* count lower than 3 log CFU/g with the microbiological method were also enumerated accordingly by the duplex PCR before and after the pre-enrichment step, respectively (Figure 5).

Taking into account the protocol used (dilution of the sample, DNA extraction, and PCR reaction), the method’s LOQ should be 4.23 log CFU/g, which is higher than the PHC. Many of these naturally contaminated samples were below the LOQ. Bland–Altman plots were constructed to assess the agreement between the microbiological method and the duplex PCR method before and after the pre-enrichment step (Figure 6). All but one of the tested samples were inside the 95% confidence interval limits (±1.96 SD) for the PCR performed before and after the pre-enrichment step. Moreover, 1.96 SD values were relatively high, ranging from to 0.75 log CFU/mL (before pre-enrichment) to 1.2 log CFU/mL (after pre-enrichment). A slight but non-significant bias towards overestimation by the duplex PCR method before the pre-enrichment step (0.04 log CFU/mL) and towards underestimation (−0.10 log CFU/mL) after pre-enrichment was observed (Figure 6). This analysis demonstrated that enumeration by the duplex PCR after the pre-enrichment step led to a lower accordance with the microbiological results than enumeration results obtained before the pre-enrichment step. However, as described above, the differences between the two methods for samples with a count below the LOQ were not systematically higher than those with *Campylobacter* amounts higher than the LOQ (Figure 6).

## 4. Discussion

Developing a rapid and reliable method to detect *Salmonella* and enumerate *Campylobacter* on broiler carcasses will help food business operators monitor these pathogens. The development of this dual-purpose method was based on the assumption that the pre-enrichment step, used for *Salmonella* multiplication in an aerobic atmosphere, would not impair *Campylobacter* enumeration by qPCR. Indeed, thermotolerant *Campylobacter* spp. are microaerophilic and generally do not grow in media incubated in an aerobic atmosphere during the initial isolation procedure [15]. According to Anis et al. (2022), a positive effect of *Salmonella* on *C. jejuni*’s survival could be observed when *C. jejuni* was co-incubated with *Salmonella* during a 16-hour incubation in peptone broth in aerobic conditions, depending on the *C. jejuni* strains and the *Salmonella* serovars. However, *C. jejuni* was unable to grow with or without *Salmonella* during this incubation [11].

Before testing whether a dual-purpose method could be optimized, it was necessary to develop a duplex PCR designed to amplify specifically the DNA from both *Salmonella* spp. and *Campylobacter* spp. while at the same time being reliable for *Campylobacter* spp. quantification. In this study, primer pairs and probes from previous studies were used, as they had already been validated. For *Campylobacter* spp. detection and quantification, the primers and probes initially described by Lund et al. (2004) were used [12]. These primers target the 16S rRNA gene from *Campylobacter* spp. The authors demonstrated good specificity regarding the thermophilic *Campylobacter* strains mainly found in poultry such as *C. jejuni*, *C. coli*, *C. lari*, and *C. upsaliensis* [12]. Moreover, Botteldoorn et al. (2008) obtained the highest specificity with these primers when they tested different primer pairs for the detection of *Campylobacter* spp. [16]. They also obtained a suitable correlation for the quantification of *Campylobacter* spp. on poultry carcass rinses between this qPCR method and the bacteriological one [16]. For *Salmonella* detection, the primers and probe described by Malorny et al. (2004) targeting the ttrRSBCA locus responsible for tetrathionate respiration were used [13]. The authors demonstrated high specificity regarding the genus *Salmonella* and the method was highly accurate compared with the traditional culture method [13]. These primers have also been used to detect *Salmonella* spp. from different matrices (veal, pork, and poultry) [17]. In this work, the results showed that it was indeed possible to amplify both *Campylobacter* and *Salmonella* DNA in the same PCR reaction. The specificity of the duplex PCR was confirmed using different in-house strains of *Campylobacter* spp. belonging to the most prevalent genotypes of *C. jejuni* present in poultry [18] and different serovars of *Salmonella* spp. The specificity of the primers/probes used in this case had previously been evaluated separately by Lund et al. (2004) [12] and Malorny et al. (2004) [13], but our results confirmed that specificity was not impaired during the duplex PCR.

The performance characteristics of the duplex PCR were evaluated. A linear relationship between the DNA input and the Cq of the duplex PCR for *Campylobacter* and *Salmonella* genomic DNA was demonstrated using the tested protocol for amplification. The LOD for *Salmonella* using the duplex PCR was 0.64 log CFU/reaction. Considering the method used in this work as a whole, from sample dilution to PCR, this would correspond to a theorical LOD of 3.64 log CFU/g. This detection level is similar to that previously described by Malorny et al. (2004) [13]. As a pre-enrichment step of 16 h is performed, it should be possible to obtain detectable levels following the recovery and multiplication of *Salmonella*. Taking into account the time taken for pre-enrichment, DNA extraction, and the reaction itself, the *Salmonella* detection result could be obtained in less than 24 h instead of 4-5 days with the EN ISO 6579-1 [9] method. The LOD and LOQ for *Campylobacter* determined in this study were 0.23 and 1.23 log CFU per reaction, respectively. Considering the method as a whole, the LOD and LOQ should theoretically be 3.23 and 4.23 log CFU/g, respectively, beyond the limit of 3 log CFU/g set by the PHC. The duplex PCR method would require a relatively high *Campylobacter* contamination of the neck skin for reliable detection and quantification when using the sample preparation described. Another study by Papic et al. (2017) reported a similar LOD and LOQ using a real-time PCR-based method [19]. By optimizing the protocol using a higher quantity of material for DNA extraction—concentrating the DNA after extraction or reducing the elution volume, for example—it may be possible to lower the LOQ. The LOQ is defined as the lowest amount or concentration of analyte that can be quantitatively determined with an acceptable level of uncertainty represented by a CV set to fall under 25% [20]. It is interesting to note that the lowest *Campylobacter* amount tested (0.23 log CFU reaction) gave a CV of 67.7%, but with 1.23 log CFU per reaction, the CV was only 4.83%, suggesting that intermediate amounts may need to be tested to determine if the LOQ could be lowered.

One objective of this work was to test whether *Campylobacter* enumeration by qPCR could be performed after the pre-enrichment step. A strong correlation was observed between *Campylobacter* enumeration by the microbiological method and the duplex PCR before and after the pre-enrichment step. This correlation was observed despite testing samples with *Campylobacter* amounts below the LOQ. Evaluating the agreement between the two methods revealed a non-significant bias of the duplex PCR toward underestimation, but a large standard deviation resulting in widely spaced limits of agreement was observed both before and after the pre-enrichment step. The differences between both methods for samples presenting counts below the LOQ were not systematically higher than those presenting *Campylobacter* amounts above the LOQ.

*Salmonella* and *Campylobacter* can be present on the same broiler carcasses [21], and Anis et al. (2022) have recently shown that *Salmonella* enrichment was not influenced by the presence of *C. jejuni* during the pre-enrichment step in peptone broth [11]. In the present work, the duplex PCR successfully detected *Salmonella* after the enrichment step in the presence of *Campylobacter* during artificial in vitro contamination and on naturally contaminated broiler neck skin samples. Twenty samples were analyzed and *Salmonella* was detected in only one sample by the EN ISO 6579-1 [9] method and was also detected by the duplex PCR after the pre-enrichment step. This result is not surprising considering the low level of contamination among broiler carcasses in France. In fact, Hue et al. (2011b) reported a prevalence of 7.52% on French poultry carcasses [22]. This result confirmed that the duplex PCR developed correctly detected *Salmonella* on broiler carcasses. Regarding *Campylobacter*, all but one of the samples were enumerated using the EN ISO 10272-2 method [10]; this is because in the one sample, the background microflora prevented us from spotting typical *Campylobacter* colonies. However, in this sample, *Campylobacter* was amplified by PCR. To explain the discrepancy of these results, several hypotheses could be made. The background microflora may have impaired *Campylobacter*’s growth, or *Campylobacter* cells may have been stressed or in a viable but non culturable (VBNC) state and did not recover in the culture medium and growth conditions. Indeed, *Campylobacter* can enter a VBNC state, and both refrigeration and oxygen concentration are reported to induce this state [23,24]. Another hypothesis could be that the contaminating *Campylobacter* spp. was not thermotolerant and was unable to grow in the conditions described by the ISO method. However, it could be detected by the duplex PCR as it targeted the 16S RNA gene in *Campylobacter*. These different hypotheses reveal drawbacks of both the microbiological and the PCR method; in several cases, the microbiological method could not enumerate *Campylobacter*, whereas the duplex PCR allowed the enumeration of non-thermotolerant *Campylobacter* that are not targeted in the PHC or the enumeration of non-viable *Campylobacter*.

When comparing agreement between the microbiological method and the duplex PCR, greater SD values and widely spaced limits of agreement were observed with the duplex PCR after the pre-enrichment step on naturally contaminated neck skin samples. It could be hypothesized that fat content released into the media during the incubation step could have impaired DNA extraction and/or amplification. Furthermore, several discrepancies were observed when comparing the results obtained with the reference microbiological method and the duplex PCR (before or after the pre-enrichment step). In particular, samples with a *Campylobacter* count around the PHC limit (3 log CFU/g) could be misclassified as over or under the PHC limit with the duplex PCR compared with the reference method. However, one sample simultaneously contaminated with both *Salmonella* and *Campylobacter* gave congruent results with both the ISO methods and the duplex PCR performed after the pre-enrichment step, demonstrating that simultaneous detection of *Salmonella* and enumeration of *Campylobacter* in a dual-purpose PCR should be possible. The ISO 10272-2:2017 microbiological enumeration of *Campylobacter* spp. is the “gold standard” method and all new methods developed should be compared to it prior to validation [10]. However, the ISO method has been previously reported to exhibit strong variance [25,26]. Indeed, due to the reproducibility limit determined during the interlaboratory validation of the method, it was concluded that results up to 3.77 log CFU/g (5900 CFU/g) on poultry skin samples do not indicate non-compliance with the PHC limit [25]. Moreover, Stingl et al. (2021) [26] recently developed an alternative qPCR method based on the detection of viable *Campylobacter* spp. in chicken meat rinses and concluded that this method was more reliable than the ISO method.

## 5. Conclusions

Encouraging results were obtained with the duplex PCR method for simultaneous *Salmonella* detection and *Campylobacter* quantification, but the major issue lies in the fact that the method’s quantification of *Campylobacter* is not yet reliable at the limit set by the PHC. The different steps of this dual-purpose PCR method each need to be optimized to fine-tune the method as a whole.

## Figures and Tables

**Figure 1 pathogens-12-00338-f001:**
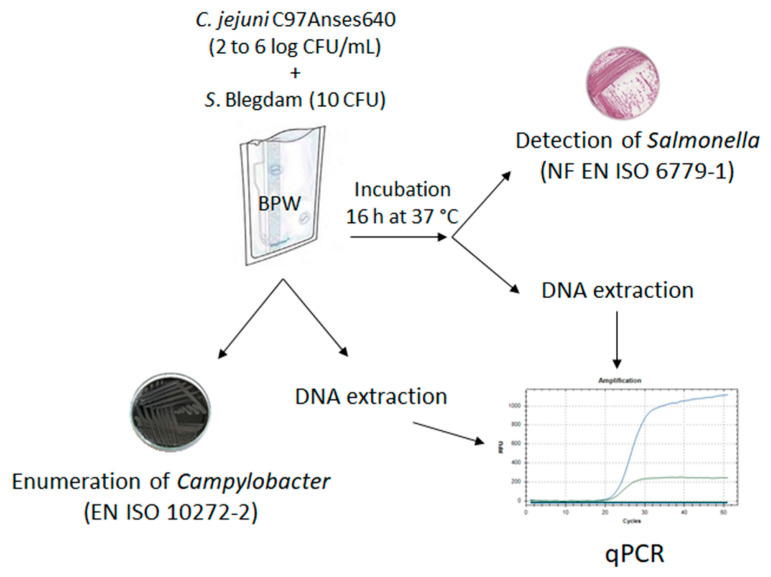
Schematic protocol used to compare the results between the duplex PCR method and the ISO methods for *Campylobacter* enumeration and *Salmonella* detection on artificially inoculated buffer peptone water (BPW).

**Figure 2 pathogens-12-00338-f002:**
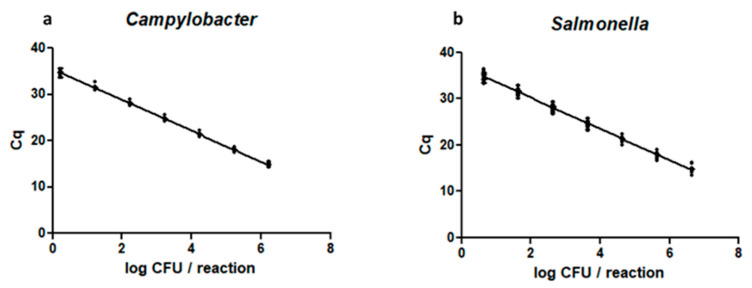
Linear relationship between the DNA input (log CFU per reaction) and the Cq of the duplex PCR for *Campylobacter* (**a**) and *Salmonella* (**b**) genomic DNA.

**Figure 3 pathogens-12-00338-f003:**
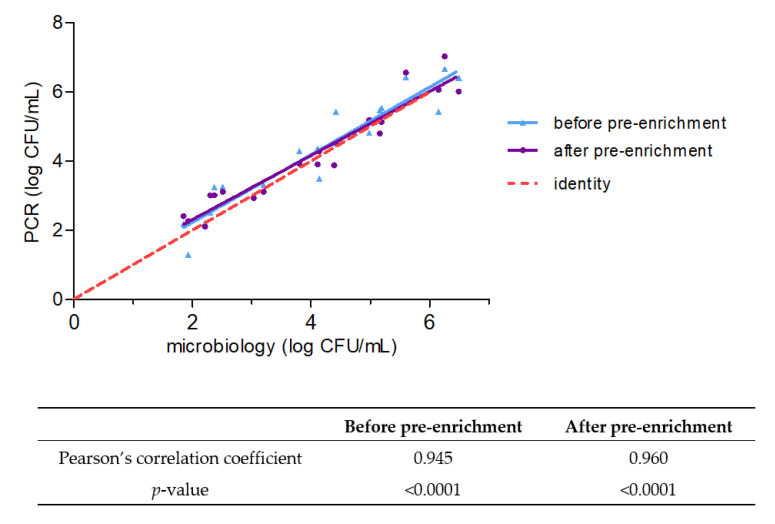
Correlation of the results obtained for *Campylobacter* spp. enumeration by the microbiological method (EN ISO 10272-2) and the duplex PCR performed before or after the pre-enrichment step.

**Figure 4 pathogens-12-00338-f004:**
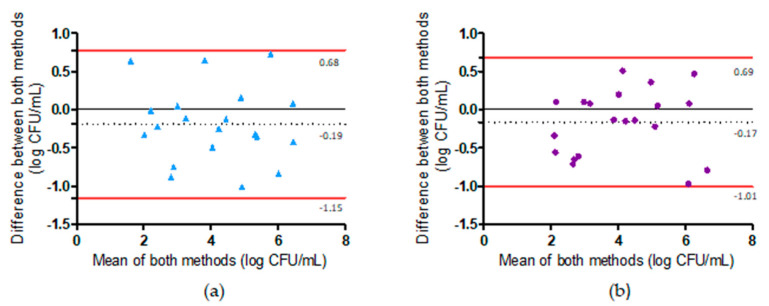
Bland–Altman analysis to evaluate the agreement of the microbiological method (ISO 10272-2) and the duplex PCR performed before or after the pre-enrichment step for *Campylobacter* enumeration in artificially contaminated in vitro samples. (**a**) Agreement between the duplex PCR when performed before the pre-enrichment step and the microbiological method. Mean bias −0.19 ± 0.49 log CFU/mL (95% confidence intervals from −1.55 to 0.78 log CFU/mL, *n* = 20). (**b**) Agreement between the duplex PCR when performed after the pre-enrichment step and the microbiological method. Mean bias −0.17 ± 0.43 log CFU/mL (95% confidence intervals from −1.01 to 0.68 log CFU/mL, *n* = 20).

**Figure 5 pathogens-12-00338-f005:**
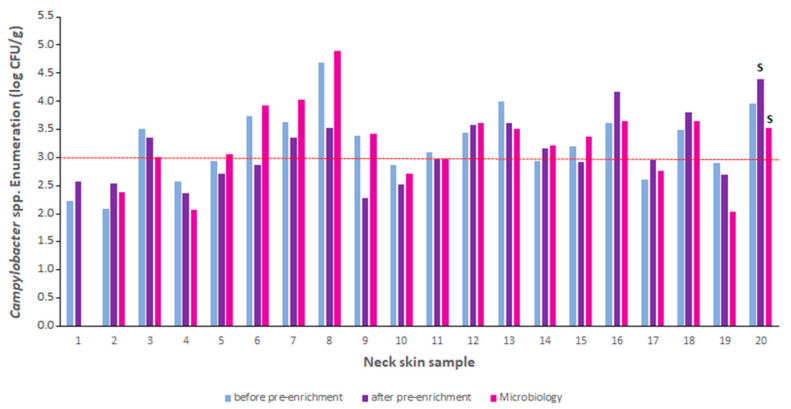
Comparison of the results obtained using the microbiological methods for *Campylobacter* enumeration (EN ISO 10272-1) and *Salmonella* detection (EN ISO 6579-1) and the duplex PCR method before and after enrichment. *Campylobacter* counts are represented by vertical bars. *Salmonella* detection is shown with an “S” above the corresponding sample and method used. The red dotted horizontal line shows the limit of 3 log CFU/g set by the *Campylobacter* PHC.

**Figure 6 pathogens-12-00338-f006:**
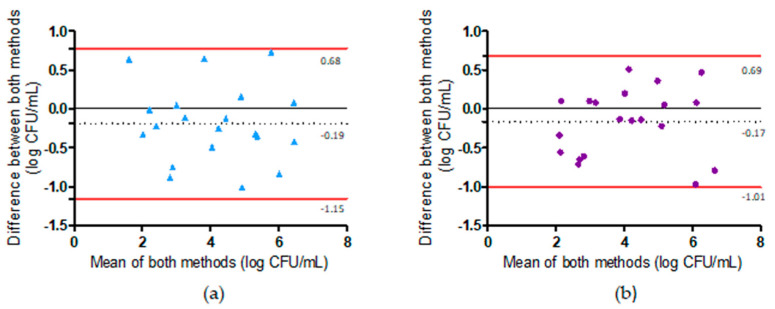
Comparison of the results obtained using the microbiological methods for *Campylobacter* enumeration (EN ISO 10272-1) and *Salmonella* detection (EN ISO 6579-1). (**a**) Agreement between the PCR when performed before the pre-enrichment step and the microbiological method. Mean bias −0.04 ± 0.35 log CFU/mL (95% confidence intervals from −0.64 to 0.73 log CFU/mL, *n* = 20). (**b**) Agreement between the PCR when performed after the pre-enrichment step and the microbiological method. Mean bias −0.10 ± 0.61 log CFU/mL (95% confidence intervals from −1.30 to 1.09 log CFU/mL, *n* = 20).

**Table 1 pathogens-12-00338-t001:** Bacterial strains used for specificity testing of the duplex PCR. All the tested strains are in-house strains except those with a CIP or ATCC number, which come from the Pasteur Institute collection or the American Type Collection, respectively. For *C. jejuni* strains, the clonal complex determined by MLST is in brackets.

*Campylobacter* Strains	*Salmonella* Strains	Other Bacterial Strains
*C. jejuni* C97Anses640	*S.* Blegdam 421	*Escherichia coli* CIP 53.126
*C. jejuni* AC0473 (ST-21)	*S.* Typhimurium S17LNR1383	*Proteus mirabilis* CIP 103181T
*C. jejuni* AC0400 (ST-45)	*S.* Enteritidis S17LNR1420	*Klebsiella pneumonia* K11RS01
*C. jejuni* AC4322 (ST-464)	*S.* Infantis S20LNR0009	*Pseudomonas aeruginosa* CIP 76.110
*C. jejuni* AC0302 (ST-206)	*S.* Hadar S20LNR0028	*Yersinia enterocolitica* CIP 81.41
*C. jejuni* AC0541 (ST-257)	*S.* Virchow S19LNR0182	*Shigella flexneri* CIP 82.48
*C. jejuni* AC0306 (ST-61)	*S.* Indiana S20LNR0422	*Staphylococcus aureus* CIP 76.25
*C. jejuni* AC0272 (ST-48)	*S.* Saintpaul S20LNR0439	*Listeria monocytogene* CIP 59.53
*C. jejuni* AC0190 (ST-353)	*S.* Derby S20LNR0321	*Enterocococus faecalis* CIP 103214
*C. jejuni* AC0484 (ST-354)	*S.* Livingstone S20LNR0708	*Rhodococcus hoaggi* ATCC 6939
*C. jejuni* AC0290 (ST-460)	*S.* Mbandaka S20LNR0056	*Citrobacter braakii* ATCC 51113
*C. jejuni* AC0332 (ST-22)	*S.* Rissen S20LNR1127	*Arcobacter butzleri* CIP 103493
*C. jejuni* AC0571 (ST-283)	*S.* Montevideo S20LNR1226	*Arcobacter skirrowi* CIP 1035588
*C. jejuni* AC0587 (ST692)	*S.* Napoli S20LNR0121	
*C. jejuni* AC0630 (ST443)	*S.* Dublin S20TA004	
*C. jejuni* AC0662 (ST-1150)	*S.* Gallinarum S19LNR0801	
*C. jejuni* C0066 (ST-1034)	*S.* Anatum S20LNR1294	
*C. jejuni* C0125 (ST-658)	*S.* Senftenberg S20LNR1352	
*C. jejuni* C0816 (ST-573)	*S.* Kedougou S20TYP002	
*C. jejuni* C0386 (ST-42)	*S.* Agona S20LNR0146	
*C. jejuni* C0398 (ST-446)	*S.* Chester S20LNR0560	
*C. jejuni* 70.2T	*S.* Newport S20LNR0763	
*C. jejuni* 103778	*S.* Kentucky S18LNR1175	
*C. coli* CIP 70.80T	*S.* Panama S20LNR1113	
*C. lari* CIP 1027221	*S.* Give S20LNR1119	
*C. fetus* C03FM1499	*S.* Venezia S20LNR1316	
*C. hyointestinalis* C12PT516		

**Table 2 pathogens-12-00338-t002:** Primers and probes used for the duplex PCR.

Target	Primer/Probe	Sequence 5′-3′	Amplicon Size (bp)	Final Concentration (nM)	Reference
*Campylobacter* spp. (16S rRNA gene)	campF2 (forward)	CACGTGCTACAATGGCATAT	108	900	[12]
	campR2 (reverse)	GGCTTCATGCTCTCGAGTT		900	
	campP2 (probe)	HEX ^2^-CAGAGAACAATCCGAACTGGGACA-BHQ1 ^3^		125	
*Salmonella* spp. (ttr ^1^ locus)	ttr6 (forward)	CTCACCAGGAGATTACAACATGG	95	100	[13]
	ttr4 (reverse)	AGCTCAGACCAAAAGTGACCATC		100	
	ttr5 (probe)	FAM ^4^-CACCGACGGCGAGACCGACTTT-BHQ1		125	

^1^ ttr: tetrathionate respiration; ^2^ HEX: hexachlorofluorescein (reporter dye); ^3^ BHQ: black hole quencher (quencher); ^4^ FAM: 6-carboxyfluorescein (reporter dye).

**Table 3 pathogens-12-00338-t003:** Performance characteristics of the duplex PCR. The mean values ± standard deviation (SD) from ten replicates (*n*) are shown.

*Campylobacter* Standard (Mean Values ± SD)	
**Slope**	−3.3 ± 0.0
Correlation coefficient (R^2^)	1.000 ± 0.000
Efficacy in %	99.7 ± 1.5
*n* Standard curves	10
***Salmonella* Standard (Mean Values ** **± SD)**	
**Slope**	−3.4 ± 0.1
Correlation coefficient (R^2^)	1.000 ± 0.000
Efficacy in %	97.7 ± 2.4
*n* Standard curves	10

**Table 4 pathogens-12-00338-t004:** Determination of the limit of detection and limit of quantification for *Campylobacter* enumeration. The expected log CFU/reaction is based on the result obtained by the microbiological method assuming no loss during DNA extraction. The coefficient of variation is defined as the ratio of the standard deviation to the mean and is expressed in %.

Expected Log CFU/Reaction	Average Observed log CFU/Reaction ± SD	Coefficient of Variation (%)
0.23	0.24 ± 0.16	67.70
1.23	1.22 ± 0.06	4.83
2.23	2.21 ± 0.05	2.14
3.23	3.25 ± 0.07	2.20
4.23	4.24 ± 0.06	1.46
5.23	5.26 ± 0.08	1.55
6.23	6.20 ± 0.10	1.59

**Table 5 pathogens-12-00338-t005:** Effect of different concentrations of *C. jejuni* Anses640 on *S.* Blegdam detection by the duplex PCR method. The threshold cycle (Cq) value is shown.

*Campylobacter* Concentration (log CFU/mL)	Cq for *Salmonella* Amplification After Enrichment (Mean ± SD)	*Salmonella* Detection by the ISO Method
6	15.58 ± 0.39	presence
5	15.51 ± 0.69	presence
4	15.42 ± 0.66	presence
3	15.00 ± 0.36	presence
2	14.93 ± 0.50	presence

## Data Availability

The data presented in is study are available on request from the corresponding author.

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
