# Peer review of "Simultaneous Detection of Salmonella spp. and Quantification of Campylobacter spp. in a Real-Time Duplex PCR: Myth or Reality?"

_pathogens, 2023, doi:10.3390/pathogens12020338_

Round 1
Reviewer 1 Report
Title of the article: Simultaneous Detection of Salmonella spp. and Quantification 2 of Campylobacter spp. in a Real-time Duplex PCR: Myth or Reality?
The following issues are to be addressed before acceptance
Introduction:
1. Line no.30-32 – rewrite the sentence as ‘Campylobacter and Salmonella spp. Are the leading bacterial pathogens causing gastroenteritis in humans. In Europe, during the year 2020, 120,946 cases of campylobacteriosis and 52,702 cases of Salmonellosis has been reported.’
2. Line 34 – include ‘ …………they remained the most frequently reported zoonotic diseases
3. Line 37-41 – rewrite the sentence as
Commission Regulation (EU) No 2073/2005 [6] has laid down the microbiological criteria for foodstuffs. In particular, there is a process hygiene criterion (PHC) on Salmonella or Campylobacter? for broiler carcasses after chilling. This regulation was amended in 2018 for Campylobacter according to Commission Regulation (EU) No 2017/1495 [7].
4. Line 54 – the reference methods used for………….
5. Line 65 – Therefore, the recommended neck skin……………….
6. Line 66 – remove the word based on
Materials and Methods:
1. Line 102 : Rewrite the sentence. You have used the same probe sequence with only change in reporter dye, so that it wont be a actual modification. Instead you may write like ‘The reporter dye has been replaced with HEX for the probe targeting the Campylobacter in order to facilitate differentiation between two pathogens in a single reaction.’
2. Section 2.4: the subheading and the content does not match. The content of this subheading may be included in section 2.2
3. In the section 2.5, the specificity check procedure may be included
4. Line 143 - 3.36 x 108 CFU/mL ……………. x 108 CFU/Ml
5. Line 144 – rewrite as ‘The genomic DNA of the standard cultures were extracted as mentioned in the section 2.2 and was serially diluted. The bacterial load of Campylobacter and Salmonella per reaction ranged from 0.23 to 6.23 log CFU and 0.64 to 6.64 log CFU, respectively.’
6. Section 2.6 – Since you have used standard cultures, is the pre-enrichment step of 16 h for Salmonella is really required? A simple cocktail of Campylobacter (2 to 6 CFU/mL) and Salmonella (10 CFU/mL) may be directly subjected for DNA extraction and duplex qPCR (for detection and quantification). There are also studies reported that even 4-6 h enrichment in Salmonella may favour its detection in molecular assay i.e. LAMP, qPCR etc., The author should look into this critically so as to reduce the time consumption.
7. The figure number in the text of section 2.6 is not matching with actual figure.
8. Line 171-172 – rewrite the sentence as ‘To evaluate the diagnostic potential of the developed duplex qPCR………………..
9. Line 175 – Figure number to be changed
10. Line 176 – rewrite as ‘25g of neck skin samples was homogenized in 250 ml of BPW [1:10 (m/v)]……….’
11. Pictorial representation of the procedure may be removed, as it duplicates the one mentioned in section 2.6 except the sampling note.
Results:
1. 3.1 subheading may be written as ‘Performance efficiency of the developed duplex qPCR’
2. Line 199 – 202 – delete it as it is repetition of statement mentioned in materials and methods section. Start the statement as ‘ The duplex realtime PCR evaluated using serial dilutions of target pathogens, Campylobacter and Salmonella showed a linear relationship between………..’
3. Mention about the specificity of the assay followed by sensitivity.
4. Delete the sentence in line 216.
5. Line 220 – correct the sentence as ‘The limit of quantification (LOQ) determined for Campylobacter is shown in the table 4’
6. Delete the line 220-223 ‘As reviewed…………………………..25%’
7. Line 223 – Delete the word ‘Regarding this definition’. Rewrite it as ‘The results presented in the table 4 revealed that the LOQ of Campylobacter is 1.23 log CFU/reaction with CoV of 4.83% which is the acceptable level. Whereas, the highest variability (67.70%) was observed in the reaction having lowest quantity of Campylobacters i.e. 0.23 log CFU/reaction.’
8. Section 3.3 – delete line 243-246
9. Section 3.4 – delete lines 255-262. Repetition of materials and methods
10. Line 312 – rewirite the sentence as ‘Only on sample (Sample no.20) was found positive both in conventional and duplex qPCR whereas Campylobacter was detected and enumerated in one sample by conventional method but in all by the developed duplex qPCR’.
11. Line 317 – statement is not clear.
12. The first paragraph in the section 3.5 may be represented in a tabular form for better understanding
13. Line 335 – Check the LOQ
Discussion:
There are few repetitions of statements in discussion section in first and second paragraph. Discussion may be improved further.
Overall suggestions:
I would like to appreciate the efforts of the authors in developing this assay. This assay may be used simultaneously detect and enumerate both the pathogens at a time. There are many such kind of assays developed earlier targeting upto 11 pathogens for their detection in food matrices. The author would have tried using a minimum pre-enrichment time for Salmonella. Did the author try any other method of DNA extraction? Because the procedures involved in DNA extraction IS time consuming including the pre-enrichment step. If so, then the entire procedure can be performed and results can be obtained in a day. Materials and methods are repeated in results section, the author should take care of it. Also maintain uniformity in mentioning the assay name: Duplex PCR/duplex qPCR/realtime duplex PCR.

Author Response
We thank the reviewer for the interest in our work and for helpful and relevant comments that will greatly improve the manuscript. We have revised the manuscript and made the suggested changes in the revised manuscript (with track change). Below is a detailed response (in blue) to the reviewers’ comments.
Title of the article: Simultaneous Detection of Salmonella spp. and Quantification 2 of Campylobacter spp. in a Real-time Duplex PCR: Myth or Reality?
The following issues are to be addressed before acceptance
Introduction:
- Line no.30-32 – rewrite the sentence as ‘Campylobacter and Salmonella spp. Are the leading bacterial pathogens causing gastroenteritis in humans. In Europe, during the year 2020, 120,946 cases of campylobacteriosis and 52,702 cases of Salmonellosis has been reported.’
Correction was made accordingly (Line 30-32)
- Line 34 – include ‘…………they remained the most frequently reported zoonotic diseases
Correction was made accordingly (Line 35)
- Line 37-41 – rewrite the sentence as
Commission Regulation (EU) No 2073/2005 [6] has laid down the microbiological criteria for foodstuffs. In particular, there is a process hygiene criterion (PHC) on Salmonella or Campylobacter? for broiler carcasses after chilling. This regulation was amended in 2018 for Campylobacter according to Commission Regulation (EU) No 2017/1495 [7].
Corrections was made accordingly (Line 39-41). Initially, the PHC set by the Commission Regulation (EU) No 2073/2005 was only for Salmonella.
- Line 54 – the reference methods usedfor………….
Correction was made accordingly (Line 56)
- Line 65 – Therefore, the recommendedneck skin……………….
Correction was made accordingly (Line 67)
- Line 66 – remove the word based on
Correction was made accordingly (Line 68)
Materials and Methods:
- Line 102 : Rewrite the sentence. You have used the same probe sequence with only change in reporter dye, so that it wont be a actual modification. Instead you may write like ‘The reporter dye has been replaced with HEX for the probe targeting the Campylobacter in order to facilitate differentiation between two pathogens in a single reaction.’
Correction was made accordingly (Line 123-125)
- Section 2.4: the subheading and the content does not match. The content of this subheading may be included in section 2.2
The content of the section 2.4 was moved in section 2.1. (Line 91-108). This section was subsequently renamed “2.1. Strains and culture conditions”. Section numbers were modified accordingly
- In the section 2.5, the specificity check procedure may be included
This section is now the section 2.4. The specificity check procedure was included as requested (Line 163-164). Consequently, this section was renamed “Evaluation of the duplex PCR’s specificity Duplex PCR and performance characteristics”.
- Line 143 - 3.36x 108 CFU/mL ……………. x 108 CFU/Ml
Correction was made accordingly (Line 170)
- Line 144 – rewrite as ‘The genomic DNA of the standard cultures were extracted as mentioned in the section 2.2 and was serially diluted. The bacterial load of Campylobacter and Salmonella per reaction ranged from 0.23 to 6.23 log CFU and 0.64 to 6.64 log CFU, respectively.’
Correction was made accordingly (Line 171-175)
- Section 2.6 – Since you have used standard cultures, is the pre-enrichment step of 16 h for Salmonellais really required? A simple cocktail of Campylobacter (2 to 6 CFU/mL) and Salmonella (10 CFU/mL) may be directly subjected for DNA extraction and duplex qPCR (for detection and quantification). There are also studies reported that even 4-6 h enrichment in Salmonella may favour its detection in molecular assay i.e. LAMP, qPCR etc., The author should look into this critically so as to reduce the time consumption.
This work was firstly designed as a proof of concept. One of the objective was to compare the results of the duplex-PCR to the results obtained with the ISO methods. We used the enrichment conditions described in the ISO 6579-1 for Salmonella, and we were able to detect 10 CFU in 250 mL after enrichment. We agree that the protocol could now be improved by adjusting several parameters to reduce the time for analysis.
- The figure number in the text of section 2.6 is not matching with actual figure.
The figure number was corrected to Figure 1
- Line 171-172 – rewrite the sentence as ‘To evaluate the diagnostic potential of the developed duplex qPCR………………..
Correction was made accordingly (Line 201)
- Line 175 – Figure number to be changed
Figure 2 was removed as suggested
- Line 176 – rewrite as ‘25g of neck skin samples was homogenized in 250 ml of BPW [1:10 (m/v)]……….’
Correction was made accordingly (Line 206-209)
- Pictorial representation of the procedure may be removed, as it duplicates the one mentioned in section 2.6 except the sampling note.
Figure 2 was removed as suggested
Results:
- 3.1 subheading may be written as ‘Performance efficiency of the developed duplex qPCR’
Subheading was corrected as suggested (Line 234)
- Line 199 – 202 – delete it as it is repetition of statement mentioned in materials and methods section. Start the statement as ‘ The duplex realtime PCR evaluated using serial dilutions of target pathogens, Campylobacter and Salmonella showed a linear relationship between………..’
Correction was made accordingly
- Mention about the specificity of the assay followed by sensitivity.
The section 3.2 was renamed “Specificity and sensitivity of the duplex PCR” and the sensitivity of the duplex PCR was moved after the mention of the specificity as suggested (Line 253-260).
- Delete the sentence in line 216.
Correction was made accordingly
- Line 220 – correct the sentence as ‘The limit of quantification (LOQ) determined for Campylobacter is shown in the table 4’
Correction was made accordingly (Line 266)
- Delete the line 220-223 ‘As reviewed…………………………..25%’
Correction was made accordingly
- Line 223 – Delete the word ‘Regarding this definition’. Rewrite it as ‘The results presented in the table 4 revealed that the LOQ of Campylobacter is 1.23 log CFU/reaction with CoV of 4.83% which is the acceptable level. Whereas, the highest variability (67.70%) was observed in the reaction having lowest quantity of Campylobacters i.e. 0.23 log CFU/reaction.’
Correction was made accordingly (Line 271-273).
- Section 3.3 – delete line 243-246
Correction was made accordingly
- Section 3.4 – delete lines 255-262. Repetition of materials and methods
Correction was made accordingly
- Line 312 – rewirite the sentence as ‘Only on sample (Sample no.20) was found positive both in conventional and duplex qPCR whereas Campylobacter was detected and enumerated in one sample by conventional method but in all by the developed duplex qPCR’.
In fact, for Salmonella detection, only one sample was found positive using the microbiological method and the duplex PCR method after enrichment. Campylobacter were enumerated in 19/20 samples with the microbiological method but in 20/20 samples with the duplex PCR method. The sentence was amended (Line 363-365).
- Line 317 – statement is not clear.
The text was amended for clarification (Line 368-373).
- The first paragraph in the section 3.5 may be represented in a tabular form for better understanding The results are obtained from Figure 5. The text was amended for clarification (Line 368-373).
- Line 335 – Check the LOQ
The LOQ was estimated for the whole method, not only the LOQ for the PCR step as described in section 3.2
Discussion:
There are few repetitions of statements in discussion section in first and second paragraph. Discussion may be improved further.
Several statements were removed (Line 417-419; Line 446-448; Line 452-454)
Overall suggestions:
I would like to appreciate the efforts of the authors in developing this assay. This assay may be used simultaneously detect and enumerate both the pathogens at a time. There are many such kind of assays developed earlier targeting upto 11 pathogens for their detection in food matrices. The author would have tried using a minimum pre-enrichment time for Salmonella. Did the author try any other method of DNA extraction? Because the procedures involved in DNA extraction IS time consuming including the pre-enrichment step. If so, then the entire procedure can be performed and results can be obtained in a day. Materials and methods are repeated in results section, the author should take care of it. Also maintain uniformity in mentioning the assay name: Duplex PCR/duplex qPCR/realtime duplex PCR.
As mentioned in the manuscript, the objective of this work to test the proof of concept that Salmonella could be detected and Campylobacter enumerated from the same sample of broiler neck skins and with a single reaction. We agree with the reviewer, several parameters remain to be tested to improve the method.
Duplex PCR was used to describe the method developed in this work throughout the manuscript
Reviewer 2 Report
Reviewer comments
General comments
The study provides information on the simultaneous detection of Salmonella and quantification of Campylobacter in single PCR run. The study is well planned, organized and the results are well presented and discussion and conclusion is in coherent to the results presented. The authors were able to identify the areas with discrepancies compared to the standard method which may make the developed assay unsuitable and suggest the areas for improvement.
Specific area comments
|
Line 110 |
Which materials used as negative and positive controls? |
|
Line 112 “………using for…..” |
“……..used for………” |
|
Line 138 “…..itsLOD……” |
“……..its LOD………” |
|
Section 2.6 |
Consider replacing “suspension” with the name of the media/broth inoculated i.e. “inoculated BPW” |
|
Line 167-168 “…..contaminated bacterial suspensions. |
Consider revising as suggested above |
|
Line 204 “Cq” |
Long form when first used |
|
Figure 3 |
Maintain uniformity, the figure label should be same that use in figure legend i.e. A for A not A for a |
|
Line 223 “…….(CV)set……” |
“………….(CV) set……” |
|
Line 225-227 “Indeed, the observed CV increased with lower concentrations; it was 67.70 % with 0.23 log CFU/reaction and only 4.83% with 1.23 log 226 CFU/reaction”. |
What does this tell us in relation to the suitability of the assay? |
|
Line 274-275 “This strong correlation was observed despite samples with Campylobacter amounts below the LOQ” |
Re-write this sentence |
|
Line 287 “A slight but non-significant……..” |
Consider revising to “A slight and non-significant……” |
|
Line 290-294 |
This results section, report results and consider shifting your opinion to discussion section |
|
Line 313 -314 “Campylobacter was enumerated in all but one sample (sample 1) with the microbiological method; this sample presented atypical colonies on selective agar plates (data not shown)” |
Not clear |
|
Line 470 “…….were obtained……” |
“…were observed…..” |

Author Response
We thank the reviewer for the interest in our work and for helpful and relevant comments that will greatly improve the manuscript. We have revised the manuscript and made the suggested changes in the revised manuscript (with track change). Below is a detailed response (in blue) to the reviewers’ comments.
|
Line 110 |
Which materials used as negative and positive controls? Correction was made accordingly (Line134) |
|
Line 112 “………using for…..” |
“……..used for………” Correction was made accordingly (Line 137) |
|
Line 138 “…..itsLOD……” |
“……..its LOD………” Correction was made accordingly (Line 165) |
|
Section 2.6 |
Consider replacing “suspension” with the name of the media/broth inoculated i.e. “inoculated BPW” Correction was made accordingly throughout the manuscript |
|
Line 167-168 “…..contaminated bacterial suspensions. |
Consider revising as suggested above Correction was made accordingly |
|
Line 204 “Cq” |
Long form when first used Correction was made accordingly (Line 242) |
|
Figure 3 |
Maintain uniformity, the figure label should be same that use in figure legend i.e. A for A not A for a Correction was made accordingly |
|
Line 223 “…….(CV)set……” |
“………….(CV) set……” Correction was made accordingly |
|
Line 225-227 “Indeed, the observed CV increased with lower concentrations; it was 67.70 % with 0.23 log CFU/reaction and only 4.83% with 1.23 log 226 CFU/reaction”. |
What does this tell us in relation to the suitability of the assay? The sentence was corrected (Line 271-273) |
|
Line 274-275 “This strong correlation was observed despite samples with Campylobacter amounts below the LOQ” |
Re-write this sentence The sentence was removed as it was already mentioned in the discussion section |
|
Line 287 “A slight but non-significant……..” |
Consider revising to “A slight and non-significant……” Correction was made accordingly (Line 335) |
|
Line 290-294 |
This results section, report results and consider shifting your opinion to discussion section The sentence was removed as it was already mentioned in the discussion section |
|
Line 313 -314 “Campylobacter was enumerated in all but one sample (sample 1) with the microbiological method; this sample presented atypical colonies on selective agar plates (data not shown)” |
Not clear In fact, for Salmonella detection, only one sample was found positive using the microbiological method and the duplex PCR method after enrichment. Campylobacter were enumerated in 19/20 samples with the microbiological method but in 20/20 samples with the duplex PCR method. The sentence was amended (Line 363-365). |
|
Line 470 “…….were obtained……” |
“…were observed…..” Correction was made accordingly (Line 524). |